# Do Think Tags Really Help LLMs Plan? A Critical Evaluation of ReAct-Style Prompting

## Abstract

The reasoning abilities of Large Language Models (LLMs) remain a topic of debate, which are critically tested in sequential decision-making problems. ReAct, a recently popular method has gained popularity for claiming to enhance LLM reasoning abilities while directly prompting them by "*interleaving reasoning trace with action execution*" in text-based planning domains such as AlfWorld and WebShop. However, given the different components of ReAct-style prompting, it remains unclear what the source of improvement in LLM performance is. In this paper, we critically examine the claims of ReAct-style prompting for sequential decision-making problems. By introducing systematic variations to the input prompt, we perform a sensitivity analysis along the original claims of ReAct. Contrary to these claims and common use-cases that utilize ReAct-style prompting, we find that the performance is minimally influenced by the interleaved reasoning trace or by the content of these generated reasoning traces. Instead, the performance of LLMs is primarily driven by the unreasonably high degree of similarity between input example tasks and queries, implicitly forcing the prompt designer to provide instance-specific examples which significantly increases the cognitive burden on the human. Our empirical results, on the same suite of domains as ReAct, show that the perceived reasoning abilities of LLMs stem from the exemplar-query similarity and approximate retrieval rather than any inherent reasoning abilities.

## 1 Introduction

Large Language Models (LLMs) have seen rapid advancements specifically in Natural Language Processing and Understanding (NLP & NLU). LLMs have unparalleled capabilities in text generation, summarization, translation, question answering to name a few. (Bubeck et al., 2023). Motivated by these capabilities of LLMs, there has also been a rush to look for other emergent abilities–especially for reasoning and planning. A popular way of enhancing LLM performance on reasoning/planning tasks has been in-context prompting or prompt-engineering (Sahoo et al., 2024) to include instructions (Giray, 2023), syntax structure (Marvin et al., 2023), criticism and plan guidance with verification (Kambhampati et al., 2024) etc. Among these approaches, ReAct (Yao et al., 2022b), presented at ICLR 2023, stands out which claims to improve LLM planning abilities through the use of reasoning traces interleaved with action execution given as plan guidance. While their original study includes multiple domains tasks such as question answering on HotPotQA and FEVER datasets (Yang et al., 2018; Thorne et al., 2018), in this paper we are particularly interested in planning tasks where ReAct claims superior performance.

In our initial experiments with ReAct for planning, we found that the system is overly dependent on a high degree of syntactic similarity between the example prompt and the query, and is extremely brittle to minor perturbations to the input prompt. For example, when provided with an explicit set of examples of *pick-and-place-object* task and asked to plan for a *pick-and-place-two-objects* task, it should be trivial to generalize the solution of the examples to the queried task. Unfortunately, even such a minor variation to the original ReAct agent setup disrupts its performance.

Given the seemingly widespread adoption of ReAct methodology (as of this writing, it has 1,408 citations), the brittleness we witnessed calls for a systematic study of the factors contributing to the performance of ReAct-based LLM Agents. Moreover, recent studies have highlighted similar case studies on the original Chain-of-Thought idea (Stechly et al., 2024a; Sprague et al., 2024). Based on

the claims of (Yao et al., 2022b), we isolate three possible reasons for the claimed performance of ReAct framework: 1) the utility of interleaving reasoning trace during action execution, 2) the utility of providing plan guidance, and, 3) the significance of example prompt provided to the the LLM.

The brittleness of ReAct becomes apparent when considering the variability in prompt designs. Depending on the domain and task, prompt designers may provide abstract guidance, task-specific instructions within the same domain, analogical examples, or global reasoning traces. ReAct's inability to robustly adapt to such variations underscores its limitations in handling diverse input prompts, thereby necessitating a closer examination of its design and implementation. In this work, we systematically evaluate the brittleness of ReAct by studying which potential factors contribute to its performance. This analysis is conducted through variations in input prompts to understand how a ReAct LLM Agent responds to (1) where the guidance is provided, (2) the different types and structure of this guidance, and finally, (3) on varying the resemblance of example prompt to the queried problem. We investigate the research questions :

**RQ1:** Does the agent performance depend on interleaving reasoning trace with action execution?
**RQ2:** How does the nature of the reasoning trace or guidance information affect the performance of LLM Agents?
**RQ3:** How does the similarity between the example ⟨problem, solution⟩and the query ⟨problem, ? ⟩, which are present in the prompt, affect LLM Agent performance?

We conduct extensive experiments on the AlfWorld and WebShop domain using various LLM Models, including GPT-3.5-turbo, GPT-3.5-instruct, GPT-4-0314(Old Variant), GPT-4-0613 (Latest Variant), GPT-4o, Claude-Opus and Llama 3.1-8b. Through our comprehensive empirical study, we answer each of the research questions above. (RQ1) We find that LLM performance in-fact improves when the reasoning trace is **not** interleaved with action execution. (RQ2) Moreover, providing weaker guidance or placebo-guidance (where the text provides no information about the task) has comparable performance to strong reasoning trace-based guidance. Answers to RQ1 and RQ2 suggest that the source of ReAct LLM agent performance is **not** the interleaving aspect or the content of the reasoning trace. Finally, in (RQ3), we see that variations to the example prompt such that it belongs to different task within the same domain, or has a different goal or plan attributes than the queried problem; causes the performance of ReAct-LLM Agent to plummet.

Our findings highlight that the benefits of ReAct-style prompting are present when prompt engineers can curate highly instance-specific examples. This may not scale for domains with a large number of problem instance classes, and it places an undue burden on prompt engineers to provide instance spe-cific examples. Finally, our experiments call into question claims of enhanced "emergent reasoning" of LLMs with prompt engineering efforts; corroborating contemporary research (Verma et al., 2024; Valmeekam et al., 2024; Stechly et al., 2024b; Ullman, 2023; Schaeffer et al., 2023; McCoy et al., 2023; Stechly et al., 2024a; Sprague et al., 2024) questioning reasoning abilities of LLMs.

For the rest of the paper, we begin with discuss the relevant literature discussing LLM prompting for planning problems in Section 2. Next, we provide the preliminaries in Section 3, and discuss our investigations of ReAct-style prompting in detail in Section 4. We present our empirical results with a detailed discussion drawing insights from our results in Section 5. Finally, we conclude the work in Section 6. An appendix with additional experiment details, results, and prompts, has also been attached, and the code will be released on acceptance.

## 2 RELATED WORK

Large Language Models have been shown to be successful in a plethora of natural language tasks (Kocoń et al., 2023; Gilardi et al., 2023; Zhu et al., 2023; Bubeck et al., 2023; Bhattacharjee et al., 2024). However, there are two schools of thought when it comes to utilizing off-the-shelf LLMs for planning and reasoning tasks. Works such as Chain of Thought, ReAct, and others that followed (Wei et al., 2022; Yao et al., 2023; Long, 2023; Yao et al., 2024; Besta et al., 2024; Fu et al., 2024; Aksitov et al., 2023), have argued about the reasoning abilities of LLMs by proposing prompting methods. On the other hand, (Valmeekam et al., 2024; Stechly et al., 2024b) have refuted these claims by showing the inability of LLMs to solve deterministic planning and classical reasoning problems.

In particular, for investigating the use of LLMs in solving decision making problems, AlfWorld (Shridhar et al., 2020) is a popular domain that was originally proposed for training text-based

```
                                    ReAct
Interact with a household to solve a task. Here are two examples.
<EXAMPLE 1 of same task as QUERY>
You are in the middle of a room ... <Task Description> Your task is to : put some spraybottle on toilet.

Act 1: think : To solve the task, I need to find and take a spraybottle, then put it on toilet.
Obs 1: OK.
Act 2: think : First, I need to find a spraybottle. A spraybottle is more likely to appear in cabinet (1-4),
        countertop (1), toilet (1), sinkbasin (1-2), garbagecan (1).
        I can check one by one, starting with cabinet 1.
Obs 2: OK.
Act 3: go to cabinet 1.
Obs 3: On the cabinet 1, you can see cloth 1, a soapboar 1, a soapbottle 1.
...
Act 6: think : Now I find a spraybottle (2), next I need to to take it.
Obs 6: OK.
...
<EXAMPLE 2 of same task as QUERY>
...
<QUERY>
Here is the task. You are in the middle of the room ... Your task is to put some soapbottle in toilet.
```

■ Interleaved Think tag
■ Reasoning Trace / Guidance
■ Example Task / Query Task

Figure 1: An example of ReAct in AlfWorld. We highlight the main components of ReAct, i.e., Interleaved reasoning and acting, the reasoning trace / plan guidance and the example and query task.

Reinforcement Learning agents. Lately, works such as ReAct, Reflexion, and their other variants (Yao et al., 2022b; Shinn et al., 2023) have argued on the prowess of LLMs' reasoning abilities on AlfWorld. Furthermore, there have been several extensions to ReAct that boost their generalization abilities across more domains including multi-modal domains (Yang et al., 2023; Castrejon et al., 2024), autonomous vehicles (Cui et al., 2024), table question answering (Zhang et al., 2023), etc. While the effectiveness of ReAct is celebrated across different areas, these works only depend on anthropomorphization of LLMs (Min et al., 2022; Peng et al., 2024) for using ReAct based prompting with no justification on the source of improvement in performance. This motivates our work in investigating the components of ReAct with respect to sequential decision-making problems and analyzing the role each component plays.

## 3 PRELIMINARIES

### 3.1 DOMAINS

**AlfWorld:** (Shridhar et al., 2020) is a synthetic text-based game built on top of a STRIPS-style PDDL domain description (Fikes & Nilsson, 1971). ReAct (Yao et al., 2022b) defines six tasks (or problem classes) within this domain namely - Put, Clean, Heat, Cool, Examine, and PutTwo. Each problem class consists of several problem instances, such as *put a spraybottle on toilet* (see Fig. 1 is an example instance of Put class. Since AlfWorld is a partially observable environment, each of these problem instances can be solved by navigating and interacting with the environment simulator via text actions. For example, this task can be solved by the following actions- `go to cabinet 2, take spraybottle 2 from cabinet 2, go to toilet 1, put spraybottle 2 in/on toilet 1`.

**WebShop:** (Yao et al., 2022a) is an online shopping website environment with 1.18M real-world products and 12K human instructions. The agent is provided with an initial human instruction (for example, "I am looking for a nightstand with drawers. It should have a nickel finish, and priced lower than $140"). The agent's task is to crawl the shopping environment using actions such as `search 'nightstand drawers'`, `choose 'white buttons'`, `back to search`, etc. For this work, we randomly sample 500 test instructions from the environment and evaluate the success rate of the agent's task completion.

### 3.2 REACT

ReAct (Yao et al., 2022b) claims to increase LLM's performance on text-based planning tasks such as AlfWorld and WebShop primarily by augmenting the original action space of the agent with a *think* action. The *think* action tag provided by ReAct is claimed to comprise of **Re**asoning + **Act**ion trace that is provided in the solution for the example problems (exemplars) as part of the prompt. During

execution, the LLM is expected to generate a *think* action tag for the queried problem instance that is semantically similar to the one provided for the examples in the prompt.

**Location of *THINK* tag** In ReAct, the integration of the *think* tag within actions serves to expand the action space. This allows the language model (LLM) agent to execute a *think* action, prompting an *'OK'* response. Through analysis of example prompts in ReAct experiments, we identify various instances of the `think` action. Typically, it appears after stating the problem instance, reiterating the task, and providing problem-specific guidance. However, the authors offer no structured guidelines for its implementation, placement, or guidance. This observation aligns with feedback from the paper's reviewers (OpenReview, 2024) citing inconsistencies in the prompting format.

**Content of *THINK* tag** In ReAct, the *think* action consistently provides the decision-making agent with success-oriented guidance for task completion. For instance, upon encountering a `spraybottle`, the prompt might include: `think:  Now I find a spraybottle (2). Next, I need to take it`. This guidance exposes forthcoming actions and sub-tasks for the agent.

**Few shot *EXAMPLE*s** In the AlfWorld domain (wihch is a PDDL domain), ReAct authors (Yao et al., 2022b) classify six problem classes or tasks: `Put, Clean, Heat, Cool, Examine, PutTwo`. Despite representing different tasks, they share the same environment dynamics and action space, allowing for very similar execution trace. For instance, a `Heat` task might involve `Putting` an item into a microwave. In ReAct experiments, authors provide two example problem-solution pairs (referred to as exemplars in our work) before querying the LLM agent with a problem instance. Authors force ReAct agent to use examples and queries belonging to the same problem class without motivating this design decision. However, the queried problem may differ in objects or locations from the exemplars.

# 4 A CRITICAL EVALUATION OF REACT-STYLE PROMPTING

We examine the claims of ReAct to understand the performance of ReAct-based LLM agents. It is crucial to assess whether ReAct's fundamental claims hold, particularly in planning. As outlined in Section 3, ReAct comprises three main components: interleaving the `think` tag with actions, plan guidance after the `think` tag, and the selection of exemplar problems for LLM prompts. We perform a sensitivity analysis by proposing alternatives along these three dimensions. The subsequent sections explore the design of exemplar prompt variations to investigate our research questions concerning the claims of ReAct. Each variation modifies the base ReAct prompt, and we use the AlfWorld domain as the running example for discussing these variations in the following sub-sections. We curate variations for WebShop domain in a similar manner. Readers can find the variations in the attached supplementary materials.

## 4.1 RQ1 : INTERLEAVING *think*ING WITH ACTING

*Does the agent performance depend on interleaving reasoning trace with action execution?*

To answer this research question, we propose collating the guidance information contained within the multiple *think* tags present in the examples of the input prompt into a single *think* tag appended after the example problem is specified. This approach can be interpreted as Chain-of-Thought (Kojima et al., 2022; Wei et al., 2022), where guidance information is generated before action execution.

**Variation 1: Exemplar-based CoT** AlfWorld is a partially observable environment where an agent can only observe objects after reaching that location. Hence, we remove specific location and object identifiers to modify the *think* actions that are originally interleaved with other actions in the environment (see RQ1 a in Figure 2). Finally, we append all the *think* actions together at the beginning of the example problem. **Intuition:** Problem-specific guidance for a sequential decision-making agent can be given step-by-step (as in ReAct) or all at once. Note, that this variation is possible since AlfWorld is not a dynamically changing environment in which case providing information on the task will not be possible.

**Variation 2: Anonymized Exemplar-CoT** We take one step further and modify the *think* tag to remove references to specific locations and objects, making it more general (see RQ1 b in Figure 2). Similarly, in WebShop we anonymize specific item options as `desired options`. **Intuition:**

```
                        ┌─────────────────────────────┐
                        │   (RQ1 a) Exemplar-CoT       │
┌───────────────────────┴─────────────────────────────┴──────────────────────┐
│ <EXAMPLE 1>                                                                  │
│ You are in the middle of a room ... <Task Description>                       │
│ Your task is to : put some spraybottle on toilet.                           │
│                                                                              │
│ Act 1: think : To solve the task, I need to find and take a spraybottle,     │
│        then put it on toilet. First, I need to find a spraybottle. A         │
│        spraybottle is more likely to appear in cabinet (1-4), countertop (1),│
│        toilet (1), sinkbasin (1-2), garbagecan (1). I can check one by one,  │
│        starting with cabinet 1.                                             │
│        Now Once I find a spraybottle (2), next I need to to take it.         │
│        Now Once I take a spraybottle (2), next I need to put it in/on        │
│        toilet (1).                                                          │
│ Obs 1: OK.                                                                   │
│ Act 2: go to cabinet 1.                                                      │
│ Obs 2: On the cabinet 1, you can see cloth 1, a soapboar 1, a soapbottle 1.  │
│ ...                                                                          │
└──────────────────────────────────────────────────────────────────────────────┘
```

```
                        ┌──────────────────────────────────────┐
                        │  (RQ1 b) Anonymized Exemplar-CoT      │
┌───────────────────────┴──────────────────────────────────────┴─────────────┐
│ <EXAMPLE 1>                                                                  │
│ You are in the middle of a room ... <Task Description>                       │
│ Your task is to : put some spraybottle on toilet.                           │
│                                                                              │
│ Act 1: think : To solve the task, I need to find and                        │
│        take a spraybottle the object, then put it on toilet the desired      │
│        location. First, I need to find a spraybottle the object. A spraybottle│
│        The object is more likely to appear in cabinet (1-4), countertop      │
│        (1),toilet (1), sinkbasin (1-2), garbagecan (1). one of the different │
│        locations. I can check one by one, starting with cabinet 1 the first  │
│        location.                                                            │
│        Now Once I find a spraybottle(2) the object, next I need to take it.  │
│        Now Once I take a spraybottle (2) the object, next I need to put it   │
│        in/on toilet (1) the desired location.                              │
│ Obs 1: OK.                                                                   │
│ Act 2: go to cabinet 1.                                                      │
│ Obs 2: On the cabinet 1, you can see cloth 1, a soapboar 1, a soapbottle 1.  │
│ ...                                                                          │
└──────────────────────────────────────────────────────────────────────────────┘
```

```
        ┌──────────────────────────┐          ┌──────────────────────────┐
        │   (RQ2 a) Failure        │          │   (RQ2 c) Ordering       │
┌───────┴──────────────────────────┴──┐  ┌────┴──────────────────────────┴──┐
│ ...                                  │  │ ...                               │
│ Act 3: open cabinet 2                │  │ Act 3: open cabinet 2             │
│ Obs 3: You open the cabinet 2. The   │  │ Obs 3: You open the cabinet 2. The│
│        cabinet 2 is open. In it, you │  │        cabinet 2 is open. In it,  │
│        see a candle 1, and a         │  │        you see a candle 1, and a  │
│        spraybottle 2.                │  │        spraybottle 2.             │
│ Act 4: think : Now I find a          │  │ Act 4: think : Now I find a       │
│        spraybottle 2. Next, I need   │  │        spraybottle 2. Next, I     │
│        to take it.                   │  │        need to take it.           │
│ Act 4: put spraybottle 2 in/on       │  │ Act 4: think : Next, I need to    │
│        toilet.                       │  │        take the spraybottle 2.    │
│ Obs 4: Nothing happens.              │  │        Now I find a spraybottle 2.│
│ ...                                  │  │ ...                               │
└──────────────────────────────────────┘  └───────────────────────────────────┘

        ┌──────────────────────────────┐    ┌──────────────────────────────┐
        │ (RQ2 b) Failure + Explanation│    │  (RQ2 d) Placebo Guidance    │
┌───────┴──────────────────────────────┴─┐ ┌┴──────────────────────────────┴──┐
│ ...                                     │ │ ...                               │
│ Act 3: open cabinet 2                   │ │ Act 3: open cabinet 2             │
│ Obs 3: You open the cabinet 2. The      │ │ Obs 3: You open the cabinet 2. The│
│        cabinet 2 is open. In it, you    │ │        cabinet 2 is open. In it,  │
│        see a candle 1, and a            │ │        you see a candle 1, and a  │
│        spraybottle 2.                   │ │        spraybottle 2.             │
│ Act 4: think : Now I find a             │ │ Act 4: think : Now I find a       │
│        spraybottle 2. Next, I need to   │ │        spraybottle 2. Next, I     │
│        take it.                         │ │        need to take it.           │
│ Act 4: put spraybottle 2 in/on toilet.  │ │ Act 4: think : Take a deep breadth│
│ Obs 4: Nothing happens.                 │ │        and work on this problem   │
│ Act 5: think : Nothing happens because  │ │        step by step.              │
│        I do not have spraybottle 2.     │ │ ...                               │
│ ...                                     │ │                                   │
└─────────────────────────────────────────┘ └───────────────────────────────────┘
```

Figure 2: Example of prompt variations considered for RQ1 and RQ2.

Exemplars can be made more general by providing abstract guidance and exploiting LLMs ability to identify necessary semantic entity relations.

## 4.2 RQ2 : PLAN GUIDANCE FOLLOWING *think* TAG

*How does the nature of the reasoning trace or guidance information affect the performance of LLM?*

ReAct claims to use reasoning trace as the guidance information following the *think* tag. For instance, in ReAct () thoughts are to (1) decompose the goal (2) track subgoal completion (3) determine the next subgoal and (4) reason via common-sense where to find and object and what to do with it. It is, however, unclear what is the motivation to use these as the reasoning trace. The potential anthropomorphization of large language models (LLMs) may suggest that their thought processes are similar to the abstract plans humans make, and that they must be prompted in the same manner. However, it is unclear why this assumption should hold true. Alternatives can be, we can prompt the LLM to reflect on past failures and provide possible explanations (hindsight-guidance) or We can substitute task-relevant guidance with placebo-guidance by using "magic incantations".

**Variation 1: Failure** From the example prompts used in ReAct, we note that none of the examples for any task consist of invalid actions. We inject two invalid actions in the execution trace : the first that attempts to execute the action pertinent to the task (such as `put spraybottle 2 in/on`

`toilet`) when not possible and, second, executes some other invalid action. We include the expected simulator response, `Nothing happens.`, when invalid actions are taken. **Intuition:** Reasoning trace can be about *what to do* such as subgoals of the future, or *what not to do* such as mistakes in hindsight. This should be weaker guidance than in base ReAct as the exemplars do not point out what to do next.

**Variation 2: Failure + Explanation** We place *think* actions after invalid actions injected in Failure Variation which consist of explanations for the failure (see RQ2 b in Figure 2). **Intuition:** We can augment pointing out mistakes in hindsight with explanations to avoid similar failures. This is stronger guidance signal than Failure, however, the exemplars still not provide information on what to do next.

**Variation 3: Guidance Ordering** LLMs are known to be susceptible to minor syntactic perturbations to inputs. We test whether it is true for guidance information given as prompt as well (see RQ2 c in Figure 2). We identify chain of subtasks in a reasoning trace $S_1 \rightarrow S_2 \cdots S_n$ and reverse it to be $S_n \rightarrow S_{n-1} \cdots S_1$. **Intuition:** LLM agent should be invariant to the syntax of reasoning trace if the semantic information is preserved. This does not change the reasoning trace from the perspective of information content.

**Variation 4: Placebo Guidance** It is unclear to what extent LLM agent uses the supposed helpful thoughts for the decision making task. In this variation we replace *think* tag guidance with a placebo thought that does not contain any task relevant information (see RQ2 c in Figure 2), but has been widely used as prompt engineering trick (Kojima et al., 2022). **Intuition:** According to claims of ReAct, we expect the performance to get worse when the guidance does not have any information useful for task success.

### 4.3 RQ3 : SIMILARITY BETWEEN *EXAMPLE*S AND *QUERY*

*How does the similarity between the example ⟨problem, solution⟩and the query ⟨problem, ? ⟩, which are present in the prompt, affect LLM Agent performance?*

RQ3 investigates the role of example similarity to the query in LLM agent's performance. Establishing problem similarity can be challenging, especially where minor variations to the problem can have varied interpretations (such as an analogy to a different task altogether). Our work explores this challenge in a systematic way. During example prompt construction, prompt designers may use synonyms to refer to objects (`Domain`), come up with examples where the agent task is the same as query but the goals are different (`Instance`), or provide optimal solutions as the examples (`Optimal`) preventing LLM to obtain information regarding exploration strategy. Furthermore, given that the domain has the same underlying action dynamics and that the tasks reuse several actions, prompt designers may choose to provide query specific example prompts (as in base `ReAct`), provide one of a different task and one of the same task (`One`), provide both examples to be of a different task (`Both`), or take an exhaustive approach and provide one example of all tasks (`All`).

**Variation 1: Synonyms - (`Domain`)** For this variation, we replace the object and location names in the example prompts with their synonyms. For example, ~~spraybottle~~ $\rightarrow$ aerosolbottle, ~~cabinet~~ $\rightarrow$ cupboard, and, ~~microwave~~ $\rightarrow$ oven. We make 36 such changes to object and location names across all the examples. Note that the object names / location are unchanged for the problem query and subsequent interaction with the simulator. **Intuition:** Exemplar guidance maybe specified with alternate synonymous object and location names. Reasoning agents should be invariant to variable name substitution for closed world dynamics such as PDDL based AlfWorld.

**Variation 2: Problem Instance-level - `Instance`** We inject instance-level changes to the examples provided in the prompts. Recall that we are updating the base ReAct's prompts, where the exemplar tasks are same as the query. We change the goal location in exemplar problem to ensure that it does not match with any of the goal locations in query problem. Moreover, we add repetitive yet futile actions in the exemplar execution trace which does not effect the solution. **Intuition:** Ensuring a different goal location in exemplar from the queried problem is a natural usecase. Moreover, exemplars may contain arbitrary exploration strategies such as action repetition (Sharma et al., 2017). By ReAct's claims, LLM agent performance should not be affected.

**Variation 3: Problem Level - `Both, One, All`** Recall that the environment dynamics for all the tasks are the same. In fact, several tasks subsume the use of our tasks such as `Heat` requires the

Table 1: Average Success % of LLM for RQ1 and RQ2 on six AlfWorld tasks.

| Model / Prompt | Act | ReAct | RQ1 | | RQ2 | | | |
|---|---|---|---|---|---|---|---|---|
| | | | CoT | Anon. CoT | Placebo | Order | Failure | Explanation |
| GPT-3.5-Turbo | 34.3 | 27.6 | 46.6 | 41 | 30 | 28.3 | 43.3 | 41.6 |
| GPT-3.5-Instruct | 44 | 50.7 | 61.9 | 50.7 | 41 | 42.5 | 47 | 44.7 |
| GPT-4-0314 (Old) | - | 23.3 | 43.3 | 33.3 | 36.6 | 30 | 50 | 36.6 |
| GPT-4-0613 (Latest) | 70.0 | 26.7 | 40.0 | 26.6 | 36.6 | 30 | 60 | 36.6 |
| Claude-Opus | 43.3 | 56.6 | 50 | 46.6 | 30 | 50 | 53.3 | 30 |

Table 2: Average Success % of LLM for RQ1 and RQ2 on WebShop tasks.

| Model / Prompt | Act | ReAct | RQ1 | | RQ2 | | |
|---|---|---|---|---|---|---|---|
| | | | CoT | Anon. CoT | Placebo | Failure | Explanation |
| GPT-3.5-Turbo | 1.12 | 1.04 | 2.20 | 1.88 | 1.52 | 3.48 | 3.48 |
| GPT-3.5-Instruct | 7.24 | 7.16 | 7.52 | 6.12 | 7.40 | 7.20 | 7.24 |
| GPT-4-0613 (Latest) | 8 | 4 | 8 | 8 | 6 | 8 | 8 |
| GPT-4o | 4.64 | 2.24 | 4.68 | 4.52 | 4.08 | 4.68 | 4.68 |
| Claude-Opus | 4 | 4 | 4 | 2 | 4 | 2 | 4 |
| LLAMA-3.1-8B | 1.44 | 3.16 | 3.28 | 3.92 | 2.04 | 1.20 | 2.16 |

agent to `Put` an food in the microwave. In general, all the tasks share a large portion of actions (such as exploring cabinets and locations, picking objects etc.). Motivated by how tight relationship of these tasks we come up with three variations. First, `One`, uses one exemplar of an arbitrarily picked task and the other exemplar of the same task as the query. Second, `Both`, uses both exemplars from an arbitrarily picked task. Finally, `All`, uses a total of six exemplars (this is the only variation where we provide more than the standard two examples as in ReAct) corresponding to each task under consideration. Remember, this includes the query task which is always present at the end in the input prompt. **Intuition:** With a very similar action execution trace (such as exploration, picking and placing objects) across tasks, and shared dynamics, LLM agent should be minimally affected by the use of exemplars of a different task.

**Variation 4: Exploration Strategy - `Optimal`** As noted before, ReAct does not explain the choice of exemplars used. An important ingredient to the exemplars is the exploration strategy used. In this variation we provide exemplars which serendipitously take the optimal actions (as if the environment were fully observable) and therefore the example plan is the shortest possible. **Intuition:** Exploration strategy exposed in exemplars (that too for the same problem task) should not impact ReAct's performance if the LLM agent is reasoning instead of retrieval (or pattern matching).

## 5 RESULTS

In the following sub-sections, we will answer our three RQs using the proposed prompt variations along three dimensions, i.e., the location of the *think* tag, the content of the *think* tag, and the similarity between exemplars and queried problems. All the variations modify the base ReAct prompts and we do not present a cross between variations unless otherwise noted. While the original ReAct experiments were carried out on PaLM (currently decommissioned), we reproduce their results with newer set of models. We use GPT-3.5-Turbo (context window size: 4096 tokens), GPT-3.5-Instruct (context window size: 16,384 tokens), GPT-4 (context window size: 8192 tokens), GPT-4o (context window size: 128,000 tokens), and Claude-Opus (context window size: 200,000 tokens), which are all newer models than those benchmarked in ReAct (Yao et al., 2022b). Note, that despite using newer models, our results shed doubts on the reproducibility and consistency across models of the original paper's results. As noted, we use the setup in (Yao et al., 2022b) for all our experiments. In AlfWorld, GPT3.5(Turbo, Instruct) results are on 134 instances across six tasks, GPT-4/Claude-Opus on 60 instances (10 for each task) due to cost considerations. In WebShop, GPT3.5(Turbo, Instruct), GPT-4o, LLAMA-3.1-8B (context window size: 128,000 tokens) results are on 500 samples, GPT-4/Claude-Opus are on 50 instances due to cost considerations. In this work, we do not aim to benchmark or analyze any single LLM's reasoning abilities on decision-making tasks, but rather intend to understand the robustness/brittleness of various LLMs with respect to different components in the ReAct-style prompting method for these tasks.

Table 3: Average Success % of LLM for RQ3 on six AlfWorld tasks. OC: Out of context limit

| Model / Prompt | Act | ReAct | RQ3 | | | | | |
|---|---|---|---|---|---|---|---|---|
| | | | **Domain** | **Instance** | **Optimal** | **All** | **One** | **Both** |
| GPT-3.5-Turbo | 34.3 | 27.6 | 1.6 | 30 | 20.1 | 32 | 28.3 | 1.6 |
| GPT-3.5-Instruct | 44 | 50.7 | 47.6 | 42.5 | 39.5 | OC | 17.9 | 5.2 |
| GPT-4-0314 (Old) | – | 23.3 | 13.3 | 23.3 | 50 | 23.3 | 16.6 | 0 |
| GPT-4-0613 (Latest) | 70.0 | 26.7 | 10.0 | 20.0 | 53.3 | 23.3 | 20 | 3.3 |
| Claude-Opus | 43.3 | 56.6 | 50 | 46.6 | 43.3 | 50 | 60 | 6.6 |

## 5.1 UTILITY OF INTERLEAVING REASONING TRACE WITH ACTION EXECUTION

Given the claims of ReAct, a practitioner's expectation is that all the rows would be `red` as the performance would drop by changing the location of the think tag. From Table 1(RQ1) note that the exemplar CoT and the anonymized exemplar CoT performs significantly better than base ReAct for all GPT-X family of models. Moreover, the performance dips slightly for Claude-Opus along these variations. From Fig. 3 (larger area represents worse performance), we observe that base ReAct consistently performs worse in most of the tasks. This refutes ReAct's first claim on the importance of interleaving reasoning trace generation with action execution. Even in the case of the Claude where there is a slight dip in performance, the models seems to be performing at reasonably high success rate which questions the importance of interleaved reasoning and action execution. We omit LLAMA-3.1-8B and GPT-4o (See B.4) for AlfWorld as they achieve zero performance over baselines and all the variations. From Table 2, we find a similar pattern: CoT and Anon. CoT variants perform closely or better than the baseline ReAct. A surprising result consistent in both the domains is the performance of `Act` baseline (where think tags are absent and actions are generated directly). `Act` baseline is weaker only for two models `GPT-3.5-Instruct, Claude-Opus` for both the domains, which further questions the utility of using ReAct style paradigm in the first place.

## 5.2 UTILITY OF GUIDANCE INFORMATION FOLLOWING *think* TAG

Recall that reasoning trace guidance pertains to the prospective actions or behaviors an agent should execute (foresight guidance). This type of guidance is more informative compared to other variations, such as hindsight guidance, which focuses on past errors without providing future solution steps, and placebo guidance, which is entirely unrelated to the task. ReAct claims that reasoning trace is crucial for LLM agent performance, which would predict a decline in performance with hindsight guidance and a collapse with placebo guidance. Therefore, a practitioner would expect all the rows to be a dark shade of `red`. In contrast, our findings in Table 1 indicate that hindsight guidance (`Failure, Explanation`) actually improves the performance of the GPT family of models. The Claude-Opus model's performance remains stable with hindsight (`Failure`) guidance and declines with placebo guidance. Figure 4 illustrates these models' performance across six AlfWorld tasks and variations, showing that the performance of LLMs either improved or remained consistent when provided with weaker or irrelevant guidance information. This refutes ReAct's claim that task-specific reasoning trace is the source of LLM agent performance. Our argument that LLM agent's performance is only slightly affected by the reasoning trace explains the indifference to ordering perturbation as well. If the LLM is not utilizing the reasoning trace for decision making, change in ordering would not affect the agent's performance. Our arguments hold for the Webshop domain as well, where all of the variants perform closely or better than the baseline ReAct. Finally, contrary to the general perception that better GPT models would improve over reasoning, we find that GPT-4-(Old)'s performance is the worst among GPT-X family further highlighting the brittleness of claims of ReAct. GPT-4-(Latest) performs similarly to GPT-4-(Old), except for the Act baseline which again shows the futility of ReAct prompting. In all our experiment settings, we note that LLMs replicate the exact steps as shown for the examples in the prompts. Hence, they do not output what ReAct claims as think tags if those tags are not present in the original prompt.

## 5.3 UTILITY OF EXEMPLAR SIMILARITY TO QUERY TASK

Intuitively, the similarity of `Domain` examples is closest with base ReAct, followed by `Instance` and `Optimal` variations. Finally, `All` contains an overload of information followed by `One` and

`Both` which has the same action space but uses different tasks as exemplars. Recall that AlfWorld being a PDDL domain has a shared environment dynamics across all tasks with upto 80% of actions shared across execution traces. While ReAct does not investigate impact of varied exemplars, given the popular usage one expects LLMs to be robust to such changes especially in a common-sense household domain. Table 3 shows the severe brittleness of ReAct based LLM agent to even minor variations (such as `Domain, Instance`). Specifically, performance of GPT-3.5-Turbo and GPT-4 plumments for `Domain`. Claude-Opus which was more robust in RQ1, RQ2, is also impacted severely by `Domain, Instance` variations. Furthermore, when we do not expose the exploration strategy and only provide Optimal exemplars, the performance of LLM agents further drops (except in GPT4).

Overloading the LLMs with more exemplars `All` does not impact its performance. We posit, this is because the query-task exemplar is still part of the large input prompt. Among the two exemplars, as provided in ReAct, when one of them is of a different task (`One`) then the performance significantly reduces for LLMs. When both of the exemplars are of a different task then the performance collapses to single digit success rates for all the models. This is a key result of this work highlighting the severe dependence of LLMs on the similarity of the exemplars to the query task. Through sensitivity analysis using our RQ3 variations we could find parts of the input (the task similarity of the exemplar with query) which is the source of ReAct performance. Essentially, the LLM is mimicking / performing approximate retrieval from the context presented to it. Moreover, our results corroborate the line of research that questions the inability of LLMs to reason or plan (Verma et al., 2024; Valmeekam et al., 2024; Stechly et al., 2024b; Ullman, 2023; Schaeffer et al., 2023; McCoy et al., 2023; Stechly et al., 2024a; Sprague et al., 2024).

The reported success-rate from the ReAct paper Yao et al. (2022b) on the WebShop domain is 40%. Due to the absence of the exact queries used in the paper, we randomly sampled queries from the WebShop dataset comprising 12K records. This approach possibly resulted in the decoupling of any relationship between the exemplars and the queries. Referring to Table 2, it is evident that the performance of the WebShop ReAct agent significantly declined, reaching single digit percentages (as well as other variants). This mirrors the trends observed in the `Both` variant of the Alfworld in Table 3, further supporting our arguments.

**Unrolling and Subtask Similarity** We perform additional experiments where the query task is to essentially repeat the task in the exemplar (`Unrolling`). For instance, in AlfWorld, the exemplar is `Put` and the query is `PutTwo` to put two objects at given location. In this case, the LLM has to unroll the given advice and repeat exemplar task execution to solve the query. The success rate of GPT-3.5-Instruct (the best performing GPT model in our experiments) drops down from 52% to 9%. Similarly, we experiment with a `Subtask Similarity` variation where the exemplar task subsumes execution of the query task. For instance, the `Heat` task requires the agent to pick and place object in the microwave (which is an instantiation of `Put` task). One would expect that `Heat` is a good exemplar for `Put`, however, the performance of GPT-3.5-instruct model goes from 18% to 0% in this case. These results further underscore the brittleness and the need for instance-specific exemplars in ReAct.

**Thought operationalization ability of LLMs** Given the free form nature of thought generation and arbitrary nature of thought (about subtask, common-sense next steps etc.), checking whether the generated thoughts are in-fact reasonable is a challenging problem. For completeness, we find that 40% of the times after generation of a *think* tag, subsequent environment action taken by the LLM was invalid (for GPT-3.5-instruct) in AlfWorld. It is much higher ( 80% for GPT-3.5-Turbo, 90% for Claude-Haiku) for weaker LLM models. This further highlights the inability of LLMs to operationalize its generated thought as also seen in (Roy et al., 2024). From manual inspection we find that the typical thoughts would enlist all possible locations as next locations to visit for most of the tasks. As demonstrated in Section 5.2, the performance of LLMs actually decreases when provided with foresight guidance, as seen with the base ReAct model. A detailed investigation into the validity of the generated reasoning traces is beyond the scope of this work and is suggested as future research.

## 5.4 DISCUSSION

In this sub-section, we aim to draw insights from our experiments across the three RQs which can be further extended to understanding the limitations of LLMs for planning problems. Specifically, we discuss a) the pitfalls of using ReAct-style prompting for planning domains which could further be exacerbated by approaches that build on top of ReAct framework, and b) scalability and generalizability issues as observed across multiple LLMs.

**Pitfalls of ReAct-Style Prompting:** Recall, that ReAct claims an improved performance for text-based planning domains, namely - AlfWorld and WebShop, where the presence of a *think* tag acts as guidance for the LLM to generate the next set of actions during the LLM-environment interaction. Through our sensitivity analysis, we dissect each component of ReAct-style prompting in a critical effort to understand the factor that leads to the observed success rates in these domains. With variations on the placement (RQ1) and content (RQ2) of the *think* tag, we eliminate it as the primary cause of any improvement. Furthermore, slight variations in exemplar tasks (RQ3) lead to a stark decline in success rate, clearly indicating the dependence of performance on the highly curated instance-specific examples by domain experts. While newer research in the art of prompting has pointed out the impact of well-curated examples, our work specifically highlights exemplar-query similarity as the cause of ReAct's performance and rejects contemporary belief that the heavy-lifting of LLM reasoning & planning is done through the `think` tag.

**Relevance of ReAct to newer LLMs:** ReAct uses the `Act` baseline in their work to showcase improvements due to the presence of the proposed *think* tag. For AlfWorld, ReAct reports 45% success rate for `Act` baseline and 71% success rate for `ReAct` prompting using the PaLM model. For WebShop, ReAct reports 30.1% success rate for `Act` baseline and 40% success rate for `ReAct` prompting on PaLM. However, we note from our results on both domains that the `Act` baseline performs much better than ReAct for several LLMs, which questions on the compatibility of ReAct to newer-age LLMs. ReAct performs worse with newer models as compared to the results they report on PaLM, which is currently decommissioned. This observation also questions the contemporary belief that such prompting strategies are generalizable throughout different LLM families, including newer models.

We re-iterate our key result, given any LLM model, the success rates plummet with our RQ3 variations showing a consistent pattern of dependence on the provided examples irrespective of the LLM. Moreover, the performance of all the LLMs remain quite high (if not better) when we vary the location and content of the `think` tags. This highlights the need for higher rigor in agentic LLM experimentation and in-depth evaluation seeking source of improvements. Finally, we highlight our previous discussion on *unrolling, subtask-similarity* (discussing the brittleness of perceived reasoning abilities of LLMs) and the inability of LLMs to perform reliable *thought operationalization* as key limitations which exist despite ReAct style prompting.

## 6 CONCLUSION

ReAct based prompt engineering methods have been claimed to improve planning abilities of Large Language Models. In this study, we critically examine ReAct along three dimensions, informed by its claims and our hypotheses regarding its performance sources. Contrary to ReAct's claims, our findings reveal that its performance is **neither** due to interleaving reasoning trace and guidance information generation with action execution, **nor** due to the specific nature of the guidance information. Instead, we identify that the true source of LLM performance in sequential decision-making tasks, such as AlfWorld, is the high degree of similarity between exemplar problems (few-shot) and the query task. We also showed that ReAct is susceptible to trivial variations in exemplar prompts (such as with the use of synonyms, or `Unrolling` and `Subtask Similarity` cases). Our findings caution against an uncritical adoption of ReAct-style frameworks for their putative abilities to enhance performance in domains requiring planning. To conclude, we believe that it will be helpful for practitioners and future works to take these results into account, particularly when designing prompts for text-based decision-making problems, and benefit from avoiding putting any efforts into constructing reasoning traces but rather selecting the right examples for subsequent problems.

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
