# OpenReview forum: "Do Think Tags Really Help LLMs Plan? A Critical Evaluation of ReAct-Style Prompting"
_ICLR.cc/2025/Conference — ICLR 2025 Conference Withdrawn Submission_

### Official Review · Reviewer_5H7H · 2024-10-25

**Soundness:** 2
**Presentation:** 2
**Contribution:** 2
**Rating:** 3
**Confidence:** 4

**Summary:**

This paper examines the impact of different components of ReAct-style prompting for planning problems. The authors propose several kinds of alterations to the ReAct prompts, and find that 1) the LLM's performance is minimally influenced by the interleaved reasoning trace or the content of the reasoning trace in the prompt; 2) the performance heavily depends on a high degree of similarity between the demonstrations and queried problems. These findings indicate that LLM's reasoning/planning abilities are more fragile than normally perceived.

**Strengths:**

This work studies a wide range of settings to understand how different components of the ReAct prompts affect the model's performance. The results are insightful for researchers and practitioners to better interpret what ReAct prompts are doing and the current LLMs' reasoning/planning performance.

**Weaknesses:**

- **Incomplete analysis for supporting the claims**. The paper does not include enough details/analyses about what the LLMs' actual generations are like and their comparison with the (altered) demonstrations. Are LLMs following the changed demonstrations when addressing new problems, or are they still doing things in the original ReAct style? For example, is it possible that the LLMs are still doing interleaved reasoning and acting, even though the demonstrations are changed to have all thought steps together in the beginning? For cases where the performance drops a lot (e.g., the Domain, Instance variations), are these new errors caused by the model's decreased reasoning abilities, or simple mistakes around the surface-form symbols? Relatedly, the authors often make claims that are somewhat ambiguous on the target. For example, in lines 416-418: "...the performance of LLMs either improved or remained consistent when provided with weaker or irrelevant guidance information. This refutes ReAct’s claim that task-specific reasoning trace is the source of LLM agent performance". Is this "task-specific reasoning trace" the ones in demonstrations or those generated by the model? The results only show that LLMs don't need such reasoning traces in the demonstrations, but the LLMs could still generate good traces during inference.

- **Results are overall not surprising**. It is well known that LLMs are usually not "learning" how to perform the task from the demonstration examples, rather, the prompt mostly provides the overall format and some key anchors such as label/action space related to the test problems to shape the model generation [1, 2]. There are prior works showing that one doesn't even need to provide these explicit demonstrations for the model to work, e.g., just saying "think step by step" could elicit CoT behaviors of LLMs. It is also well-known that providing examples that are more similar to the queried problems brings better performance, and many prior efforts on demonstration selection are exactly about closing the gap between the demonstrations and queries, e.g., [3, 4]. LLMs, or ML models more broadly, generally suffer from distributional shifts which is one of the open research problems. Reporting this in some specific tasks/settings is not very significant in my view.

--

Citations
- [1] Min et al. Rethinking the Role of Demonstrations: What Makes In-Context Learning Work? EMNLP-22.
- [2] Wang et al. Towards Understanding Chain-of-Thought Prompting: An Empirical Study of What Matters. ACL-23.
- [3] Liu et al. What Makes Good In-Context Examples for GPT-3? DeeLIO-22.
- [4] Rubin et al. Learning To Retrieve Prompts for In-Context Learning. NAACL-22.

**Questions:**

See Weaknesses.

---

> ### Author Response · Authors · 2024-11-21
> **Rebuttal**
>
> We thank the reviewer for their thoughtful comments and feedback, and that they find our study effective, well-structured, and the motivation valuable with respect to future research in LLM prompting methods. We have tried addressing each of the concerns below:
>
> $$\textbf{Experiment analysis (W1a):}$$ We thank the reviewer for this insightful question. In all our experiment settings, we note that LLMs replicate the exact steps as shown for the examples in the prompts. Hence, they do NOT output what ReAct claims as think tags if those tags are not present in the original prompt. We agree that this needs to be clarified clearly in the paper’s results, and thus, we have updated our Results section for lines 425-427 to further emphasize on this point.
>
> $$\textbf{Failure analysis (W1b):}$$ We would like to highlight that these failures are due to LLMs being unable to correctly reason possibly due to the mismatch with the prompt examples as shown by our results in Table 3, and clearly not due to simpler syntactic mistakes. This is also true in the case even if one of the two prompt examples is the same as the problem query (see variation descriptions in Section 5.3).
>
> $$\textbf{Conclusion and Takeaways (W2):}$$ In this work, we primarily intended to focus on the supposed usefulness of think tags, i.e., interleaved reasoning traces in multi-step text-based decision making problems. Our hypothesis behind the analysis was to give ReAct the benefit of doubt assuming that it is the reasoning trace (its content and location in the prompt) that leads to increased LLM performance on decision-making domains such as AlfWorld and WebShop. To recall, we note from Table 3 (RQ3) that changing the examples in the prompt drops the performance across multiple LLM models in the AlfWorld task, which is completely opposite to the case when we modify the location and content of the think tag in Table 1 (RQ1) and Table 2 (RQ2).  While our results re-iterate on other findings regarding the role of examples in the few-shot settings, we wanted to systematically study and show how the reasoning traces, which is the primary claim behind ReAct and all the follow-up works that build on ReAct [1-6], is of practically no use and only leads to requiring prompt engineers include these reasoning traces in the examples.
>
> To conclude, we believe that it will be helpful for future works to take these results into account, particularly when designing prompts for text-based decision-making problems, and benefit from avoiding putting any efforts into constructing reasoning traces but rather select the right examples for subsequent problems. We have included this point in our Conclusion section in lines 536-539.
>
> [1]Yao Yao, Zuchao Li, and Hai Zhao. Beyond chain-of-thought, effective graph-of-thought reasoning in large language models. arXiv preprint arXiv:2305.16582, 2023.
>
> [2]Noah Shinn, Beck Labash, and Ashwin Gopinath. Reflexion: an autonomous agent with dynamic memory and self-reflection. arXiv preprint arXiv:2303.11366, 2023.
>
> [3] Zhengyuan Yang, Linjie Li, Jianfeng Wang, Kevin Lin, Ehsan Azarnasab, Faisal Ahmed, Zicheng Liu, Ce Liu, Michael Zeng, and Lijuan Wang. Mm-react: Prompting chatgpt for multimodal reasoning and action. arXiv preprint arXiv:2303.11381, 2023.
>
> [4] Lluis Castrejon, Thomas Mensink, Howard Zhou, Vittorio Ferrari, Andre Araujo, and Jasper Uijlings. Hammr: Hierarchical multimodal react agents for generic vqa. arXiv preprint arXiv:2404.05465, 2024.
>
> [5] Can Cui, Yunsheng Ma, Xu Cao, Wenqian Ye, and Ziran Wang. Receive, reason, and react: Drive as you say, with large language models in autonomous vehicles. IEEE Intelligent Transportation Systems Magazine, 2024.
>
> [6] Yunjia Zhang, Jordan Henkel, Avrilia Floratou, Joyce Cahoon, Shaleen Deep, and Jignesh M Patel. Reactable: Enhancing react for table question answering. arXiv preprint arXiv:2310.00815, 2023.

---

> > ### Comment · Reviewer_5H7H · 2024-11-24
> >
> > Thank you for the response. While it resolves some of my concerns, my main assessment still holds - the findings in this paper are not surprising/enlightening enough from my view given prior studies in similar directions, and hence I will maintain my score.

---

### Official Review · Reviewer_7upD · 2024-10-25

**Soundness:** 2
**Presentation:** 3
**Contribution:** 2
**Rating:** 5
**Confidence:** 4

**Summary:**

This paper examines how effective ReAct-style prompting is for improving Large Language Models (LLMs) in decision-making tasks. While ReAct is known for boosting LLM reasoning by combining reasoning and actions, this study questions where these improvements really come from. The authors conduct experiments in planning tasks in AlfWorld and WebShop to analyze the effects of reasoning traces, plan guidance, and how similar examples are to the queries.

Contributions: The paper tests ReAct's claims and finds that LLM performance doesn't significantly improve by manipulating reasoning with action execution. The results show that the main reason for LLM performance is the high similarity between example tasks and queries, rather than reasoning abilities from reasoning traces. A detailed sensitivity analysis shows that LLM performance drops when the example-query similarity is reduced, highlighting ReAct's fragility. The study challenges the belief that LLMs develop reasoning abilities through ReAct-style prompting, showing that success is mostly due to pattern matching or retrieving similar examples.

**Strengths:**

Originality: The paper offers a new look at ReAct-style prompting by questioning if interleaving reasoning with actions truly improves LLM performance. It highlights example-query similarity as the main factor driving success, offering a different perspective.

Quality: The paper is well-structured, with clear experiments tested across different LLM models and domains. The use of varied tests and detailed analysis strengthens the reliability of the findings. The overall experiment setups are convincing and comprehensive.

Clarity: The paper is easy to understand and well-organized. Proposed ideas are reasonable, and tables and figures effectively explain the results, making the findings accessible to a wide audience.

Significance: The paper raises important questions about ReAct-style prompting, showing that example-query similarity may play a bigger role than reasoning in LLM performance. This insight could help guide future research and improve LLM prompting methods.

**Weaknesses:**

While the paper offers important insights into the limitations of ReAct-style prompting, it doesn’t fully address whether these findings apply across different scenarios and domains. The study focuses on specific tasks in AlfWorld and WebShop, but it’s unclear how generalizable the results are to other environments or more complex tasks. For example, would the same reliance on example-query similarity hold in tasks with more diverse or less structured action spaces? The lack of broader applicability raises concerns about the scalability of the conclusions, making it hard to know if the findings can be generalized to all situations where ReAct prompting is used.

The study uses black-box LLMs without fully clarifying what matters more for performance—the ReAct-style prompting itself or the inherent capabilities of the LLMs. Since LLMs differ in architecture and training, it's difficult to separate the effects of the prompt from the underlying model’s abilities. This makes it unclear whether the performance issues are caused by the prompting technique or limitations within the models themselves. Ideally, more transparency in how the LLMs are interacting with the prompts and what exactly drives the observed behavior would help strengthen the conclusions.

The paper also lacks a strong technical contribution beyond critiquing ReAct. There is no obvious novel method or solution offered to address the identified weaknesses in ReAct-style prompting. While finding flaws in prompting techniques is important, the absence of a proposed solution limits the paper’s impact. Readers are left with an understanding of what is wrong but without a clear path to improve or resolve these issues. A more balanced approach would include recommendations or a framework for improving prompting techniques, which would provide more concrete value to readers.

**Questions:**

Regarding scalability: Could you clarify how your findings apply to more complex environments or tasks beyond AlfWorld and WebShop? For instance, would the same dependency on example-query similarity hold in domains with less structured actions or greater variability?

Regarding justification of approaches: In your experiments, how much do you attribute the performance outcomes to the prompting method versus the underlying capabilities of the LLMs themselves?

Regarding workflow completeness: Did you explore how the size of the context window or the amount of training data provided to the LLMs impacts performance in your scenarios? While the analysis of ReAct’s limitations is thorough, do you have suggestions or possible remedies for reducing the reliance on example-query similarity?

---

> ### Author Response · Authors · 2024-11-21
> **Rebuttal**
>
> We thank the reviewer for their thoughtful comments and feedback, and that they find our study effective, well-structured, and the motivation valuable with respect to future research in LLM prompting methods. We have tried addressing each of the concerns below:
>
> $$\textbf{Generalizability of results to other tasks (W1/Q1):}$$ We would like to first highlight that we show experiments on both decision-making benchmarks that were originally used in ReAct. Furthermore, to recall the working of ReAct using the example of Put task (e.g., put some bottle on toilet) in the AlfWorld domain, the LLM needs to be provided the exact same action space via the examples to solve a subsequent Put task problem. As we show in our results for changing the examples in the prompt in Table 3, the LLM performance significantly diminishes. Note that there are cases where the examples can be from the Heat task (where the agent has to put some object in the microwave and heat it) that can be expected to work for a Put task problem (where the agent only has to put some object. The reason is that the action space for the Heat task already subsumes the action space required for solving the Put task, and yet, the LLM’s sensitivity is revealed by a significant drop in performance due to lack of generalizability. In this case, our argument is further pronounced that the LLM’s performance has little to no correlation with the interleaved reasoning traces (the example for Heat task contained the interleaved reasoning trace or the think tag which is similar to the think tag used for a Put task example).
>
> $$\textbf{Approach justification/LLM performance issues (W2/Q2):}$$ We do include LLAMA-3.1-8B results for our RQ1 and RQ2 results on Webshop in Table 2, and observe a similar trend in results as we see in the black-box models from the GPT and Claude family. We agree that while these models may have different architectures and abilities, we tried establishing baseline ReAct results for each of these models and analyzed the gain or drop in performance across our different prompt variations. In this work, we do not aim to benchmark or analyze any single LLM’s reasoning abilities on decision-making tasks (to understand its limitations as rightly pointed by the reviewer), but rather intend to understand the robustness/brittleness of various LLMs with respect to different components in the ReAct-style prompting method for these tasks. Taking the reviewer’s suggestion, we have further updated our paper in lines 374-377 to reflect this point clearly for the readers.
>
> $$\textbf{Takeaways and workflow completeness (W3/Q3):}$$ In this work, we primarily intended to focus on the supposed usefulness of think tags, i.e., interleaved reasoning traces in multi-step text-based decision making problems. Our hypothesis behind the analysis was to give ReAct the benefit of doubt assuming that it is the reasoning trace (its content and location in the prompt) that leads to increased LLM performance on decision-making domains such as AlfWorld and WebShop. To recall, we note from Table 3 (RQ3) that changing the examples in the prompt let alone drops the performance across multiple LLM models in the AlfWorld task, which is completely opposite to the case when we modify the location and content of the think tag in Table 1 (RQ1) and Table 2 (RQ2).  While our results re-iterate on other findings regarding the role of examples in the few-shot settings, we wanted to systematically study and show how the reasoning traces, which is the primary claim behind ReAct and all the follow-up works that build on ReAct (see lines 126-130), is of practically no use and only leads to requiring prompt engineers include these reasoning traces in the examples.
> Regarding the different prompt and context window size, we do include the experiments (results shown in Table 3 for the column ‘All’) where we incorporate one example each from all AlfWorld tasks, thereby increasing the prompt length significantly as compared to the other settings. To include the point on the size of the context window clearly in our experiment section, we have now shown the context window sizes for each of the LLMs that we use for our results in lines 366-374.
>
> To conclude, we believe that it will be helpful for practitioners and future works to take these results into account, particularly when designing prompts for text-based decision-making problems, and benefit from avoiding putting any efforts into constructing reasoning traces but rather select the right examples for subsequent problems. We have included this point in our Conclusion section in lines 536-539.

---

> > ### Comment · Reviewer_7upD · 2024-11-25
> >
> > I greatly appreciate the authors' detailed and prompt responses. However, after thoroughly reviewing their responses and considering the concerns raised by other reviewers, I have decided to maintain my current score.

---

### Official Review · Reviewer_C2ur · 2024-11-04

**Soundness:** 2
**Presentation:** 1
**Contribution:** 2
**Rating:** 3
**Confidence:** 4

**Summary:**

This paper explores the impact of different components of ReAct-style prompting on the final outcomes, showing that both interleaved reasoning traces and generated reasoning traces have minimal influence on results. In contrast, the similarity between examples has a much more significant effect.

**Strengths:**

1. This paper provides a valuable exploration of how ReAct functions in decision-making tasks.

2. I highly appreciate the motivation for existing work, which holds great importance for the research community.

**Weaknesses:**

1. The writing in the paper is difficult to follow. Prompts take up too much space in each paragraph, and some words are struck through without clear explanations. The font size in the figures is too small. I am confused about the names "GPT-3.5-instruct" and "GPT-3.5-Turbo" as the authors refer to "gpt-3.5-turbo-0125" and "gpt-3.5-turbo-instruct". I also struggled to understand what "Act" is in the tables 1 as it's hard to find the introduction of baselines. Overall, the paper is not ready for publication.

2. The examples provided, such as finding an item, do not seem to have sufficient changes in the environment. The authors suggest that giving LLMs a complete action plan at the outset (e.g., if A happens, do X; if B happens, do Y) is feasible, as shown in Figure 2. I disagree as many situations could change the environment, making it impossible to provide all potential scenarios upfront.

3. Regarding RQ1 and its two variants, I didn't find any intrinsic difference between these variants and React. I feel that the variants are just human-rewritten versions of React.  A more reasonable comparison would be to directly present the user’s query to the model and let it generate a similar plan prompt. The current setup, where the authors design prompts based on potential behaviors, seems unfair.

4. The conclusion that the "similarity between the example and the query" is important feels rather obvious. Since LLMs rarely encounter agentic structure data in their training, the significance of in-context learning is naturally high. This conclusion lacks sufficient insight for me.

**Questions:**

1. In Table 1, the explanation for GPT-3.5 outperforming GPT-4o is “highlighting the brittleness of claims of ReAct.” I don’t quite understand this rationale, as it seems to imply that weaker prompting leads to more significant performance drops in stronger models. Shouldn’t stronger LLMs be more robust to suboptimal prompting? Could you provide some examples to clarify this counterintuitive result?

2. The Webshop dataset shows a low accuracy. I believe analyzing such low accuracy is not that meaningful. Other papers report much higher figures. Can you explain the reason?

---

> ### Author Response · Authors · 2024-11-21
> **Rebuttal (1/2)**
>
> We thank the reviewer for their thoughtful comments and feedback, and that they find our study effective and the motivation valuable with respect to the ongoing research. We have tried addressing each of the concerns below:
>
> $$\textbf{Prompts’ space usage and terminology (W1):}$$ We noted the reviewer’s concerns regarding the difficulty in following some parts of the paper, and have tried addressing the suggested changes for improved readability and easier understanding. Our intention behind including the important prompt changes in Section 4.1 was to provide an easy-to-reference example for each prompt variation. Based on the reviewer’s suggestion, we have updated the respective paragraphs (see blue text in Section 4.1 and Section 4.2) and refer the readers to Figure 2 which includes examples for all the variations in one place. Furthermore, gpt-3.5-turbo/gpt-3.5-turbo-0125 and gpt-3.5-turbo-instruct/gpt-3.5-instruct were used interchangeably in the paper. We included a description of the ‘Act’ baseline (originally used as a baseline in ReAct paper) in line 401 in our paper.
>
> $$\textbf{Clarification on environment dynamics (W2):}$$  We would first like to clarify, based on the original paper on AlfWorld [1] that the environment is not dynamic, but partially observable which means that the agent may only find out if an object is present in a location or not after reaching that location. Moreover, as we note from our results in Table 1 (RQ1) and Table 2 (RQ2), the idea of having these reasoning traces for a couple of examples in the prompt pushes the LLM to explore the wrong rooms (as it is merely trying to replicate a similar flow of actions as shown in the example prompt). This further emphasizes our argument that these traces are not helpful. Coming back to the reviewer’s claim on the difficulty of providing all the information upfront, we surely would not be able to construct similar prompt variants for environments which are dynamic (for example, a modified AlfWorld). We have further included this as an additional comment in our paper in lines 211-213.
>
> $$\textbf{Clarification on RQ1 variants (W3):}$$ The primary motivation behind our work is to point out the reliance on human-written prompts in ReAct. While we agree with the reviewer’s point that the LLM should generate a prompt given a user query, all the prompts used in ReAct originally require human effort to design prompts with examples that can potentially help the LLM agent to replicate the behavior on a new query problem. Our sensitivity analysis is focused on revealing this exact behavior, such that it can be easily understood which elements in the prompt indeed help with the downstream task.
>
> $$\textbf{Conclusion and takeaways (W4):}$$ In this work, we primarily intended to focus on the supposed usefulness of think tags, i.e., interleaved reasoning traces in multi-step text-based decision making problems. Our hypothesis behind the analysis was to give ReAct the benefit of doubt assuming that it is the reasoning trace (its content and location in the prompt) that leads to increased LLM performance on decision-making domains such as AlfWorld and WebShop. To recall, we note from Table 3 (RQ3) that changing the examples in the prompt let alone drops the performance across multiple LLM models in the AlfWorld task, which is completely opposite to the case when we modify the location and content of the think tag in Table 1 (RQ1) and Table 2 (RQ2).  While our results re-iterate on other findings regarding the role of examples in the few-shot settings, we wanted to systematically study and show how the reasoning traces, which is the primary claim behind ReAct and all the follow-up works that build on ReAct (see lines 126-130), is of practically no use and only leads to requiring prompt engineers include these reasoning traces in the examples.

---

> > ### Author Response · Authors · 2024-11-21
> > **Rebuttal (2/2)**
> >
> > We address the reviewer’s questions below:
> >
> > $$\textbf{Robustness to suboptimal prompting (Q1):}$$ We thank the reviewer for bringing up this point where newer and larger (in terms of trained parameters) models are expected to be more robust to prompting strategies and that the results obtained on smaller/weaker models should be further improved. We believe that the reviewer refers to line 422-425 in the paper where we mention that GPT-4-0314 (Old), and not GPT-4o, performs the worst in the GPT-X family. To our surprise too, ReAct’s performance is not consistent with this expectation thereby raising further concerns on its scalability and generalizability. Given that this work has gained a significant number of eyes to build upon, we believe it is an important observation. To recall, ReAct originally showed all results only on the PaLM model (currently decommissioned) and not on any other LLM. Hence, in an effort to understand the working of ReAct while giving it all the possible benefits of doubt, we tested with multiple LLM models (including the most recent OpenAI and Claude models accessible at the time of writing this paper).
> >
> > $$\textbf{WebShop results (Q2):}$$ Firstly, we incorporate results and the same sensitivity analysis as we did on AlfWorld, for the sake of completeness to strengthen our argument regarding the study on ReAct for decision-making domains. Since ReAct originally showed results on these two domains and that these two domains have been relatively popular for studying text-based decision-making domains, we wanted to highlight that our results are consistent across both domains. Secondly, the reason behind low accuracies in the WebShop is possibly due to two reasons: A) we do not have access to the exact test set used in the original ReAct paper (which shows 40% performance on WebShop) as the authors did not make that public, we had to randomly sample test problems from the WebShop dataset comprising 12,000 problems (line 452-457 in the paper). B) As we mention above in our response to Q1, ReAct originally showed all results only on the PaLM model (currently decommissioned) and not on any other LLM. Hence, the lack of scalability and robustness in the original approach across different LLM models could have also resulted in this diminished performance.
> >
> > [1] Mohit Shridhar, Xingdi Yuan, Marc-Alexandre Côté, Yonatan Bisk, Adam Trischler, and Matthew Hausknecht. Alfworld: Aligning text and embodied environments for interactive learning. arXiv preprint arXiv:2010.03768, 2020.

---

> > > ### Comment · Reviewer_C2ur · 2024-11-26
> > >
> > > Thank you for the author's response. It addressed some of my concerns. However:
> > >
> > > 1. The authors still did not provide a reasonable explanation for Q1. We observed that GPT-4 is worse than GPT-3, which is interesting and my question is about the reason behind this observation. However, the reply is only about the "importance"
> > >  rather than the "reason".
> > >
> > > 2. Regarding Q2, if the results from the original paper cannot be reproduced, alternative reasonable approaches should be taken, e.g., contacting the authors, rather than presenting results with significant discrepancies. This makes it difficult for me to determine whether the differences arise from errors in your own implementation. Furthermore, I do not think replacing PALM with GPT would result in such a large performance gap.
> > >
> > > 3. The importance of "in-context example similarity" has already been extensively discussed and studied in the community. I recommend that the authors focus on analyzing why the "interleaved reasoning trace" is not effective in future versions.
> > >
> > > Therefore, I will maintain my current score.

---

### Official Review · Reviewer_GqdU · 2024-11-04

**Soundness:** 3
**Presentation:** 3
**Contribution:** 2
**Rating:** 5
**Confidence:** 4

**Summary:**

This paper studies into what makes ReAct-type of prompting works. It studies into 3 factors: (1) where the guidance is provided, (2) the different types and structure of this guidance (3) on varying the resemblance of example prompt to the queried problem.

**Strengths:**

It conducts a very detailed sensitivity study into the effectiveness of ReAct prompting.

**Weaknesses:**

Studies on prompt engineering and what parts of prompt work for final result has been studied for a long time. Various research on why and what types of intermediate thinking chains/in-context learning can work (such as "Rethinking the Role of Demonstrations") and on what models they work have been studied by many papers. However, no such papers are cited or discussed.

Many of the observational conclusions such as similarity to in-context examples and uselessness of thinking trace are also discussed by various papers, and thus renders this paper's conclusion not so exciting.

**Questions:**

N/A

---

> ### Author Response · Authors · 2024-11-21
> **Rebuttal**
>
> We thank the reviewer for their thoughtful comments and feedback, and that they find our study detailed and effective in analyzing ReAct. We have tried addressing each of the concerns below:
>
> $$\textbf{Citations to ICL literature (W1):}$$ We do cite some of the relevant papers on ICL in our paper in lines 102-104 in Section 2 when talking about the different prompting strategies, and have also included the paper mentioned by the reviewer in lines 131-132. We would be happy to include any other specific references that the reviewer suggests can be helpful for our work.
>
> $$\textbf{Uselessness of think tags (W2):}$$ In this work, we primarily intended to focus on the supposed usefulness of think tags, i.e., interleaved reasoning traces in multi-step text-based decision making problems. Our hypothesis behind the analysis was to give ReAct the benefit of doubt assuming that it is the reasoning trace (its content and location in the prompt) that leads to increased LLM performance on decision-making domains such as AlfWorld and WebShop. To recall, we note from Table 3 (RQ3) that changing the examples in the prompt let alone drops the performance across multiple LLM models in the AlfWorld task, which is completely opposite to the case when we modify the location and content of the think tag in Table 1 (RQ1) and Table 2 (RQ2).  While our results re-iterate on other findings regarding the role of examples in the few-shot settings, we wanted to systematically study and show how the reasoning traces, which is the primary claim behind ReAct and all the follow-up works that build on ReAct (see lines 102-104), are of practically no use and only lead to requiring prompt engineers include these reasoning traces in the examples.

---

> > ### Comment · Reviewer_GqdU · 2024-11-24
> > **Thank you for the rebuttal**
> >
> > Thank you for the rebuttal. However, I didn't find the findings in the paper especially interesting or enlightening. Thus I will maintain my score.

---

### Author Response · Authors · 2024-11-21
**Rebuttal**

We thank all the reviewers for their thoughtful reviews. We are gratified to note that all the reviewers acknowledge the fact that our systematic analysis raises significant questions about the claims in the ReAct paper. Some of the reviewers expressed reservations about the fact that we only point out the flaws in the claims made in the ReAct paper, and don’t ourselves provide solutions to rectify those flaws. We offer two rebuttals on this: first ReAct is not just any paper–but a rather influential paper (currently 1739 citations) whose claims have largely been accepted at face value. There are very recent papers [eg. 1-6] that repeat ReACT claims and claim to further extend them. So pointing out inherent flaws in ReAct is, we believe, an important addition to the science of LLM prompting. Second, the kind of claims ReAct makes are such that there is no actual way to  make them true, as has perhaps been shown by recent work evaluating the planning (in)capabilities of LLMs [7, 8]. We hope the reviewers take this view into consideration in evaluating the strength of our contribution.


[1]Yao Yao, Zuchao Li, and Hai Zhao. Beyond chain-of-thought, effective graph-of-thought reasoning in large language models. arXiv preprint arXiv:2305.16582, 2023.

[2]Noah Shinn, Beck Labash, and Ashwin Gopinath. Reflexion: an autonomous agent with dynamic memory and self-reflection. arXiv preprint arXiv:2303.11366, 2023.

[3] Zhengyuan Yang, Linjie Li, Jianfeng Wang, Kevin Lin, Ehsan Azarnasab, Faisal Ahmed, Zicheng Liu, Ce Liu, Michael Zeng, and Lijuan Wang. Mm-react: Prompting chatgpt for multimodal reasoning and action. arXiv preprint arXiv:2303.11381, 2023.

[4] Lluis Castrejon, Thomas Mensink, Howard Zhou, Vittorio Ferrari, Andre Araujo, and Jasper Uijlings. Hammr: Hierarchical multimodal react agents for generic vqa. arXiv preprint arXiv:2404.05465, 2024.

[5] Can Cui, Yunsheng Ma, Xu Cao, Wenqian Ye, and Ziran Wang. Receive, reason, and react: Drive as you say, with large language models in autonomous vehicles. IEEE Intelligent Transportation Systems Magazine, 2024.

[6] Yunjia Zhang, Jordan Henkel, Avrilia Floratou, Joyce Cahoon, Shaleen Deep, and Jignesh M Patel. Reactable: Enhancing react for table question answering. arXiv preprint arXiv:2310.00815, 2023.

[7] Karthik Valmeekam, Matthew Marquez, Sarath Sreedharan, and Subbarao Kambhampati. On the planning abilities of large language models-a critical investigation. Advances in Neural Information Processing Systems, 36, 2024.

[8] Kaya Stechly, Karthik Valmeekam, and Subbarao Kambhampati. Chain of thoughtlessness: An analysis of cot in planning. arXiv preprint arXiv:2405.04776, 2024

---

### Note · Authors · 2024-12-05

I have read and agree with the venue's withdrawal policy on behalf of myself and my co-authors.